# Flexible Cu_3_(HHTP)_2_ MOF Membranes for Gas Sensing Application at Room Temperature

**DOI:** 10.3390/nano12060913

**Published:** 2022-03-10

**Authors:** Ashraf Ali, Husam H. D. AlTakroori, Yaser E. Greish, Ahmed Alzamly, Lamia A. Siddig, Naser Qamhieh, Saleh T. Mahmoud

**Affiliations:** 1Department of Physics, United Arab Emirates University, Al-Ain 15551, United Arab Emirates; ashraf_ali@uaeu.ac.ae (A.A.); 202070073@uaeu.ac.ae (H.H.D.A.); nqamhieh@uaeu.ac.ae (N.Q.); 2Department of Chemistry, United Arab Emirates University, Al-Ain 15551, United Arab Emirates; y.afifi@uaeu.ac.ae (Y.E.G.); ahmed.alzamly@uaeu.ac.ae (A.A.); 201050663@uaeu.ac.ae (L.A.S.); 3Department of Ceramics, National Research Centre, Cairo 68824, Egypt

**Keywords:** mixed matrix membranes, Metal-Organic Framework, H_2_S gas sensing, Cu_3_(HHTP)_2_-MOF

## Abstract

Mixed matrix membranes (MMMs), possessing high porosity, have received extensive attention for gas sensing applications. However, those with high flexibility and significant sensitivity are rare. In this work, we report on the fabrication of a novel membrane, using Cu_3_(HHTP)_2_ MOF (Cu-MOF) embedded in a polymer matrix. A solution comprising a homogenous suspension of poly-vinyl alcohol (PVA) and ionic liquid (IL), and Cu-MOF solid particles, was cast onto a petri dish to obtain a flexible membrane (215 μm in thickness). The sensor membrane (Cu-MOF/PVA/IL), characterized for its structure and morphology, was assessed for its performance in sensing against various test gases. A detection limit of 1 ppm at 23 °C (room temperature) for H_2_S was achieved, with a response time of 12 s. Moreover, (Cu-MOF/PVA/IL) sensor exhibited excellent repeatability, long-term stability, and selectivity towards H_2_S gas. The other characteristics of the (Cu-MOF/PVA/IL) sensor include high flexibility, low cost, low-power consumption, and easy fabrication technique, which nominate this sensor as a potential candidate for use in practical industrial applications.

## 1. Introduction

Chemical gases that are released into the atmosphere pose a grave threat to society at large. Monitoring these threats in real time is considered as an area of research that requires constant update, in terms of the materials involved. Materials that possess designable structures, high porosity, high surface area and desirable physicochemical properties, such as Metal–Organic Frameworks (MOFs), have been attracting appreciable attention, with a plethora of applications [1,2,3], including gas sensing [4,5,6]. The gases that are being released, as a result of direct and/or indirect human actions [7,8,9,10,11,12], result in non-lethal effects to humans, whereas others, when exposed to even a few parts per million (ppm), result in various effects, drifting from respiratory problems to inescapable death [12,13].

Among lethal gases, hydrogen sulfide (H_2_S), being flammable, colorless, extremely toxic and highly corrosive, poses a grave threat to the environment. It should be mentioned that small doses, ca 500–1000 ppm, have been shown to be lethal [14]. Humans can detect concentrations as low as 0.1 ppm, whereas the sewage air usually has up to 28 ppm of H_2_S [15]. H_2_S is also known to be heavier than air, which allows it to settle in enclosed spaces, making them a dangerous environment. The requirement of developing reliable sensors to monitor these threats in real time has always been an area of continuous research and development.

A recent review by Chen et al. [16] outlines the advances in the materials for the detection of diverse gases that pose a threat. Among these materials, MOFs that can be synthesized using a bottom-up approach, have been deployed as chemical sensors [17,18,19]. This is attributed to their unique properties that can be tailored based on the requisite application. Another appreciable property of MOF materials is the high surface area to volume ratio, which heightens the chances of interaction between the sensor materials and the test gas molecules, leading to high sensitivity [20]. This property is also important to decrease the amount of material used to fabricate the membrane, which also helps to reduce the thickness of the flexible sensing layer [21,22,23]. Another vital property is the stability of the sensing layer and its selectivity to detect the target gas, among other vapor phase materials. Usually, changes in the sensor’s mechanical, photophysical or electrical properties are monitored. The magnitude of these changes depends on the concentration of the analyte, its electrical properties, such as the ability of accepting or donating electrons, and the mechanical changes induced that are due to the stress or strain developed during the sensing process. Various sensors have been developed based on these characteristics. However, it has been shown that there is always a chance for continuous improvement of the overall sensor performance.

Campbell et al. demonstrated that conductive Cu-MOF can be deployed for a sub-ppm level of ammonia vapor detection [24]. Successively, the same group demonstrated that Cu-MOF could discriminate between an array of volatile organic compounds (VOCs) based on their functional groups [25]. Smith M K et al. supported the previous work by fabricating Cu-MOF films onto graphitic electrodes, patterned on polymer films for the detection of ammonia at sub-ppm level [26]. Other gases, such as acetone vapor and NO_2,_ have also been investigated for sensing, using MOF materials [27,28]. Cu-MOF has also been used as photocatalysts [29] and supercapacitors [30] for a wider range of applications. There have been a variety of device fabrication methods employed, such as coating MOF on electrodes, using solvated “paste” [31], drop-casting [24,25], solvent-free mechanical abrasion [25], and in-situ film growth [26].

To the best of our knowledge, a composite membrane, in which Cu-MOF microparticles are blended with a polymer matrix, has not been utilized for gas detection, so far. Furthermore, blending CuO nanoparticles with a PVA mixed matrix membrane was shown to detect as low as 10 ppm of H_2_S gas at 80 °C [32]. The incorporation of the MOF particles into the polymer matrix has enhanced its sensing capability towards H_2_S gas. Hence, in this study, a flexible membrane is fabricated by incorporating various concentrations of Cu-MOF microparticles into a conducting polymer matrix, comprising PVA and IL (glycerol), denoted by Cu-MOF/PVA/IL. The results presented in this work are for the optimum concentration of the Cu-MOF blended with the PVA/IL matrix membrane that supports a detection limit of 1 ppm for H_2_S gas at 23 °C (RT).

## 2. Materials and Methods

### 2.1. Materials

Chemicals used in this study included copper (II) nitrate trihydrate (Cu(NO_3_)_2_·3H_2_O), ethanol (EtOH), aqueous ammonia (NH_4_OH), 2,3,6,7,10,11-hexahydroxytriphenylene hydrate (H_6_hhtp). All chemicals were purchased from Sigma-Aldrich and were used without purification. Distilled water (DW) used throughout the study was produced in the lab using the Milli-Q system (Elix Technology, Billerica, MA, USA).

### 2.2. Synthesis of Cu_3_(HHTP)_2_ MOF

Cu_3_(HHTP)_2_ MOF was prepared following a previously reported procedure [23]. In a Pyrex tube, 2,3,6,7,10,11-hexahydroxytriphenylene hydrate (0.30 mmol, 1 eq.) was added to 8.4 mL of DW (solution A). In a separate beaker Cu(NO_3_)_2_·3H_2_O (0.53 mmol, 1.75 eq.) was dissolved in 2 mL of DW and 30 eq. of conc. NH_4_OH (solution B). Solution B was added in a dropwise manner to solution A and placed in an oven at 80 °C for 24 h. The resultant black powder was collected using centrifugation and was washed twice with ethanol. Finally, the obtained Cu_3_(HHTP)_2_ was soaked in ethanol for 24 h and was dried in a vacuum oven at 80 °C for 24 h.

### 2.3. Fabrication of the Cu_3_(HHTP)_2_–MOF/PVA/IL Membranes

A PVA stock solution was prepared by dissolving 5 g of PVA granules in 100 mL of DW and placed on a stirrer at 70 °C at 800 rpm until a clear and homogenous solution was obtained [32,33,34]. A pre-calculated weight of 1.25 × 10^−3^ g of Cu-MOF was suspended in each ml of DW and was stirred for 15 min at 1100 rpm to attain a homogenous dispersion. The polymer membrane serves as a host for the MOF microparticles. The electrical conductivity of a PVA membrane can be controlled by doping with a suitable ionic liquid such as sorbitol, 1-methyl-3-n-decyl-imidazolium bromide, and glycerol [35,36,37]. Ionic liquids (ILs) are well known for their good ionic conductivity that is attained at room temperature. Mainly, ILs serve as electrolytes and diffusion barriers, and they have low values of vapor pressure and low toxicity. Moreover, ILs are suitable to be used for the fabrication of electrochemical sensors as they are environmentally friendly [32,33]. It has been shown that glycerol-IL can be used effectively to control the conductivity of PVA membranes [37,38], thus, it will be utilized in this work. A volume of 5 vol% of IL (glycerol) was mixed in a 20 mL of the PVA stock solution, which was also doped with Cu-MOF that was suspended in 2.5 mL of DW at 70 °C. The mixture was then taken and casted onto a petri dish and was dried in an oven at 70 °C for 16 h. The resultant membrane was found to be flexible, as shown in Figure 1, with a thickness of 215 μm.

### 2.4. Characterization

Cu_3_(HHTP)_2_ MOF powder was analyzed for its structural and morphological characteristics. A Rigaku, MiniFlex 600-C instrument, Austin, TX, USA, with a Cu *Kα* X-ray at a scan rate of 1°/min and a scan range of 2–60° was used for powder X-ray diffraction (PXRD) studies. A Thermo Scientific Quattro S—SEM instrument (Waltham, MA, USA) was used at an operating voltage of 15 kV for morphological studies of the Cu_3_(HHTP)_2_ MOF powders, as well as the composite membranes. Fourier transform infrared (FT-IR) spectra were recorded using a Thermo Nicolet, NEXUS, 470 FT-IR instrument, Ramsey, MN, USA. A KBr disk method was used and a scan range of 400–4000 cm^−1^ was employed. TGA plots were recorded using a TGA-Q500 instrument (TA Instruments, New Castle, DE, USA) within a temperature range of 30–800 °C with the ramp rate of 20 °C/min.

### 2.5. Sensor Fabrication and H_2_S Gas Sensing Tests

The device fabrication was accomplished by sandwiching a 1 × 1 cm^2^ piece of the membrane between Cu metal plate and stainless steel mesh (H_2_S resistant) with a mesh size of 200 μm as outlined in our previous study [38]. Bronkhorst mass flow controllers (MFC) were employed to dilute the test gas using synthetic air and deploying the mixture into the test chamber. A non-humid atmosphere was maintained in the test chamber at room temperature (23 °C) inside a fume hood. The gas sensing response was recorded using a Keithley Instruments source measurement unit (KI236) applying a constant bias voltage of 2.5 V between the electrodes. Labview software was used to interface and record the response of the sensor, which was recorded by measuring the electrical current signal as a function of time and different gas concentrations.

## 3. Results and Discussion

### 3.1. Structural and Morphological Characterization of Cu-MOF and Cu-MOF/PVA/IL Membrane

Figure 2A shows a comparison of the powder XRD pattern of the Cu-MOF powder with the simulated pattern, using the lattice parameters from previously reported studies [39]. The pattern was simulated using the highscore plus software package, using the said lattice parameters. The Cu-MOF pattern matches well with that reported in the literature, where peaks denoting the XRD diffraction of the 200, 210, 220, 420 and 303 planes were observed. The structural orientation of the Cu-MOF, indicating the linkages between the Cu^2+^ oxo clusters and the HHTP linker molecules, is shown in Figure 2B. The Cu-MOF exhibits the hierarchical graphene-like 2D structure, which was previously supported by the work of Hoppe et al. [23]. These morphological characteristics of the Cu-MOF crystallites were further confirmed in Figure 3. Cu-MOF crystallites appear as 2D platelets and flakes, with an average thickness of 150 nm. The selected area of the Energy-Dispersive X-ray spectroscopy (EDX) pattern of the Cu-MOF crystallites, shown in Figure 3B, indicates the phase purity of the as-prepared Cu-MOF crystallites. Figure 3C shows the N_2_-isotherm of the as-prepared Cu-MOF crystallites, where a type-IV adsorption was observed, denoting the presence of mesoporous structures, which can be attributed to the highly porous 2D morphology of the Cu-MOF platelets. It should be mentioned that the Brunauer-Emmett-Teller (BET) surface area was calculated to be 279.997 m^2^/g, which confirms the high porosity of the 2D platelets of the as-prepared Cu-MOF crystallites.

The TGA thermograms of the composite membrane and the Cu-MOF-free PVA-IL matrix were recorded from room temperature up to 800 °C, as shown in Figure 4A, where several thermal events were observed. An initial weight loss was observed around 81.6 °C in both materials, and is attributed to the evaporation of physically adsorbed water of humidity. This was followed by a second two-step event, with a first peak at 188.9 °C in both samples, while the second peak was observed at 225.5 °C for the MOF-free matrix and 215.1 °C for the composite membrane. These peaks showed a similar extent of weight loss and denote the evaporation of the remaining organic solvents [40]. The third thermal event was more pronounced and was observed at 276.5 °C for the MOF-free film, and is related to the breakdown of the PVA polymeric chains. The presence of the Cu-MOF in the composite membrane resulted in a delay in this event, where it was observed at 337.9 °C. This is attributed to the extensive removal of the organic content of the composite membrane (MOF linkers; pyromellitic acid, and PVA/IL matrix) [41]. Moreover, the shift in this event could also be used to indicate a chemical stabilization of the polymeric matrix, through extensive H-bonding formation between the highly functionalized Cu-MOF and the PVA/IL matrix. This has been further confirmed by the presence of a multi-peak thermal event of the MOF-free matrix and the presence of a single event for the composite membrane, as shown in the fourth thermal event. The multi-step weight loss could be related to the successive degradation of the polymeric chains in the MOF-free matrix. On the other hand, the absence of these multi-steps in the thermogram of the composite membrane confirms the stabilization of the matrix by the MOF particles, hence, their degradation takes place as a single thermal event. A final thermal event was observed as a plateau, at a temperature above 550 °C, where there was no appreciable weight loss denoting the formation of stable inorganic phases [41]. The profile of the recorded TGA matches well with the literature for Cu-MOF [28,29], also confirming the incorporation of Cu-MOF in the membrane. The FT-IR spectra of the pure Cu-MOF, the PVA/IL matrix and the composite membrane are shown in Figure 4B, where all bands are identified. The similarity between the FT-IR spectra of the PVA/IL matrix and the composite membrane is attributed to the presence of a small proportion of the Cu-MOF, which is below the detection limit of FT-IR as a technique [41].

Figure 5A,B shows the SEM micrograph and EDX elemental analysis pattern of the Cu-MOF/PVA/IL composite membrane. A uniform distribution of the Cu-MOF 2D platelets was observed in the SEM micrograph of the composite membrane. The weak absorption of Cu in the EDX pattern of the composite membrane could be attributed to the coverage of the Cu-MOF platelets within the polymeric matrix, in addition to the presence of Cu-MOF in a small proportion, as compared with the PVA/IL matrix.

### 3.2. Gas Sensing Performance

Figure 6A records the response of the fabricated Cu_3_(HHTP)_2_ sensor at RT to H_2_S gas exposure, at various concentrations, with respect to time. Figure 6B shows the plot of the recorded currents as a function of gas concentration. No response was observed to the gas in the absence of the Cu-MOF and it can be inferred that the recorded response is due to the incorporation of the Cu-MOF into the polymer matrix.

The data show an increase in the measured current signal when the gas concentration increased. The chamber was flushed with synthetic air to exfoliate the adsorbed gas molecules from the sensing membrane before exposures to the next cycle of target gas. It was noted that the current values gradually decreased to their initial readings, demonstrating the reversibility of the sensor. The plot shows a current response in the micro ampere region for 1 ppm and, by extrapolating the plot, it can be simulated that the sensor can detect at sub-ppm levels as well. The response of the sensor has been compared to that available in the literature and is summarized in Table 1.

The sensor response (*S*%) is measured as the difference between the resistance of the sensor in synthetic air (*R_a_*) and resistance measured in target gases (*R_g_*), as shown in Equation (1).
(1)$$S\;\left(\%\right)=\frac{R_{a}}{R_{g}}\times 100$$

An amount of 100 ppm of H_2_S gas was flown to evaluate the reproducibility and long-term stability of the sensor. The test gas was introduced into the chamber for five cycles, with synthetic air flushing between each cycle. Figure 7A displays the repeatability of the sensor, showing excellent response for 100 ppm of gas. Figure 7B shows the stability over a period of 21 days, and it can be seen that the response of the sensor is in the region of 96–99%, with negligible error bars.

The sensor was further investigated for its selectivity by exposing it to 100 ppm of H_2_S, H_2_, CO and C_2_H_4_ gases. Figure 8 records the sensor response to the 100 ppm of the gases at room temperature. It can be observed that the sensor exhibits relativity low response to H_2_, CO and C_2_H_4_ gases in comparison to H_2_S gas. Another notable parameter is the sensor’s response time, which is defined as the time taken by the sensor to reach 90% of its detection efficiency. The response time of the sensor was averaged at 12 s for 100 ppm H_2_S, with the detection response of ~96–99%.

### 3.3. Gas Sensing Mechanism

The polymeric matrix of the Cu-MOF/PVA/IL membrane is hydrophilic in nature, with –OH groups aligned along the chains. The IL is hydrophilic in nature with 3–OH groups along each IL molecule. In the presence of the Cu–MOF2D crystallites, the hydrophilic endings of the HHTP ionized linker contribute to the formation of a network, with an extensive H-bonding along all participating molecules, as illustrated in Figure 9. Upon exposure to the highly acidic H_2_S molecules, they contribute to the H-bonded network of molecules, through the attraction of the H_2_S–H atoms to the high electron density, along all involved O atoms of the PVA, IL and the Cu-MOF molecules. Hence, the transfer of charged ions, protons, along the composite membrane is facilitated. Moreover, the highly conjugated H_6_hhtph linker, shown in Figure 9, enhances the conductivity among various molecules within the composite membrane. In addition, the Cu–MOF, used in the current study, is a highly porous graphene-like structure, with an average porosity of 1.55 nm [23]. Compared with an average size of a typical H_2_S molecule of 0.36 nm, the diffusion and transportation of the H_2_S molecules within the MOF molecules are, therefore, highly enhanced. It is, therefore, the synergistic effect of the highly connected molecules of the proposed nanocomposite membrane that explains its high efficiency and sensitivity toward H_2_S gas. The high porosity of the MOF component of the composite membrane was previously proven to contribute to the passage of H_2_S molecules within the composite membrane [38]. The current findings support the contribution of the highly porous MOF structures in the construction of a gas sensor assembly. Moreover, it is believed that the homogeneous distribution of the graphene-like 2D Cu–MOF crystallites within the composite membrane explains the enhanced sensitivity towards H_2_S molecules, as compared with PVA/IL membranes.

## 4. Conclusions

In Summary, MOF–Polymer mixed matrix flexible membranes were successfully prepared from Cu_3_(HHTP)_2_ microparticles and PVA polymer, blended with an ionic liquid for H_2_S gas sensing application. The as-synthesized Cu–MOF exhibits a highly porous graphene-like structure, with high interconnected porosity, on blending with the PVA/IL matrix. They serve as diffusion and transportation pathways for the H_2_S gas molecules within the sensing membranes, exhibiting high detection ability towards H_2_S, among other test gases. The results revealed high sensitivity and selectivity of the sensor towards H_2_S gas, in addition to excellent reproducibility and long-term stability. Furthermore, the sensor operates at room temperature and does not require any heating element, which significantly reduces the power consumption and fabrication costs. The sensor, with its detection capability of 1 ppm and fast response time of 12 s, has proficient industrial applications for air quality control applications.

## Figures and Tables

**Figure 1 nanomaterials-12-00913-f001:**
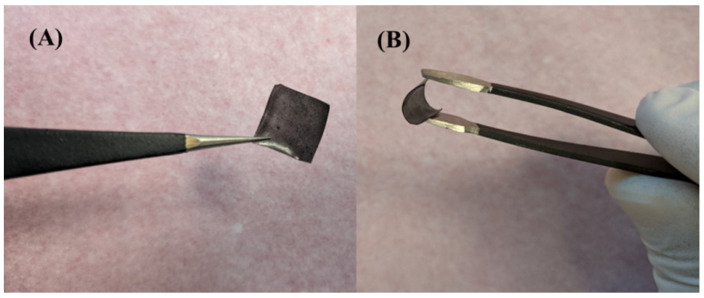
(**A**) A typical 1 × 1 cm^2^ Cu-MOF/PVA/IL membrane, (**B**) A demonstration of the high-flexibility of the membrane.

**Figure 2 nanomaterials-12-00913-f002:**
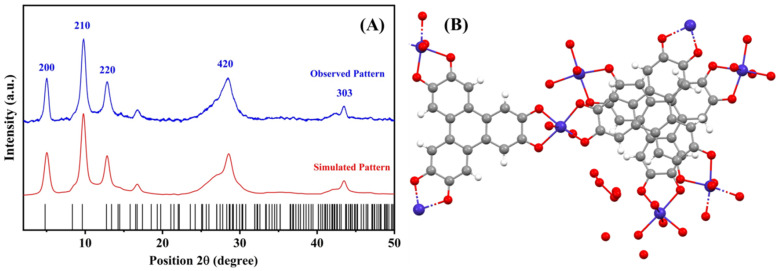
(**A**) PXRD pattern of the observed and simulated Cu-MOF powder, (**B**) Structural information of the 2D Cu-MOF.

**Figure 3 nanomaterials-12-00913-f003:**
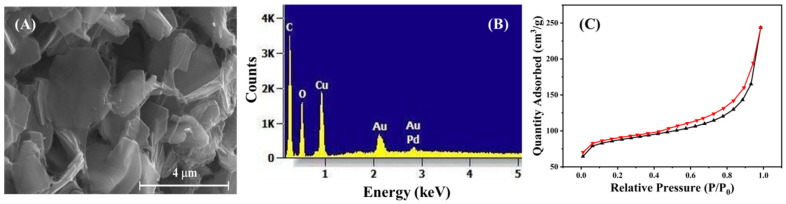
(**A**) SEM micrograph, (**B**) EDX elemental analysis pattern (**C**) N_2_-adsorption hysteresis of the as-prepared Cu-MOF powder.

**Figure 4 nanomaterials-12-00913-f004:**
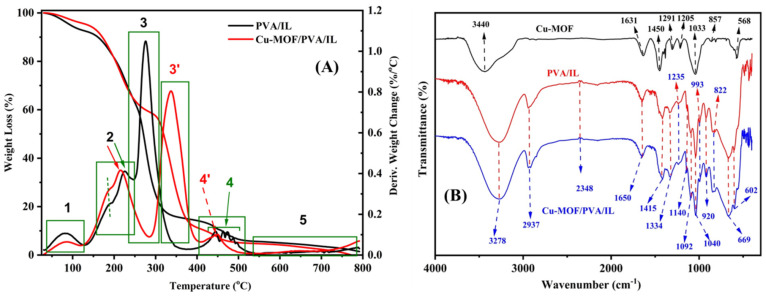
(**A**) TGA thermograms of PVA/IL and Cu-MOF/PVA/IL membranes (**B**) FTIR spectra of the as-prepared Cu-MOF powder, fabricated PVA/IL, and Cu-MOF/PVA/IL membranes.

**Figure 5 nanomaterials-12-00913-f005:**
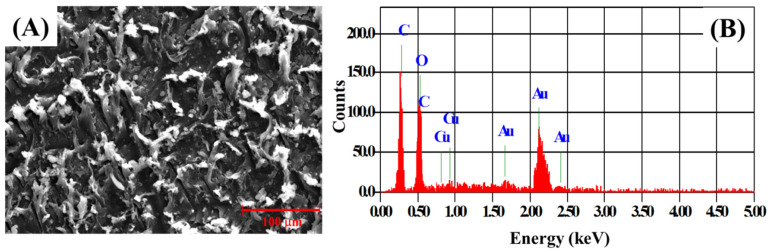
(**A**) SEM micrograph and (**B**) EDX pattern of the Cu-MOF/PVA/IL membrane.

**Figure 6 nanomaterials-12-00913-f006:**
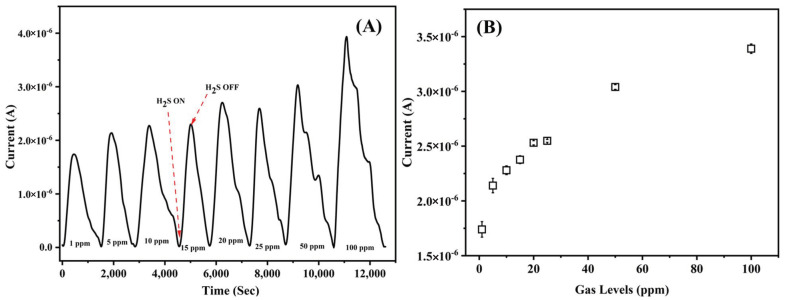
(**A**) Current response of the Cu-MOF/PVA/IL membrane with respect to time and H_2_S concentration measured at room temperature. (**B**) Response of the sensor for corresponding gas concentrations.

**Figure 7 nanomaterials-12-00913-f007:**
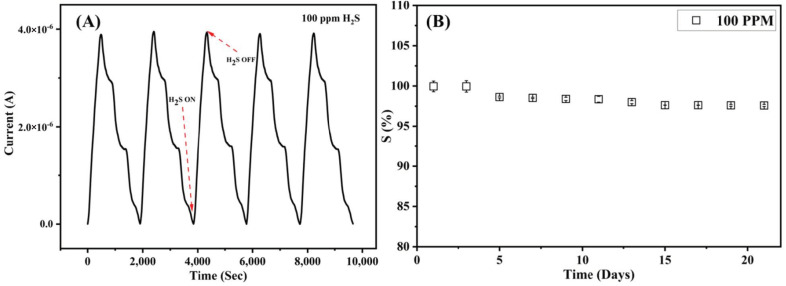
(**A**) Reproducibility of the Cu-MOF/PVA/IL membrane at RT. (**B**) Long-term stability performance of the Cu-MOF/PVA/IL sensor at RT for 21 days.

**Figure 8 nanomaterials-12-00913-f008:**
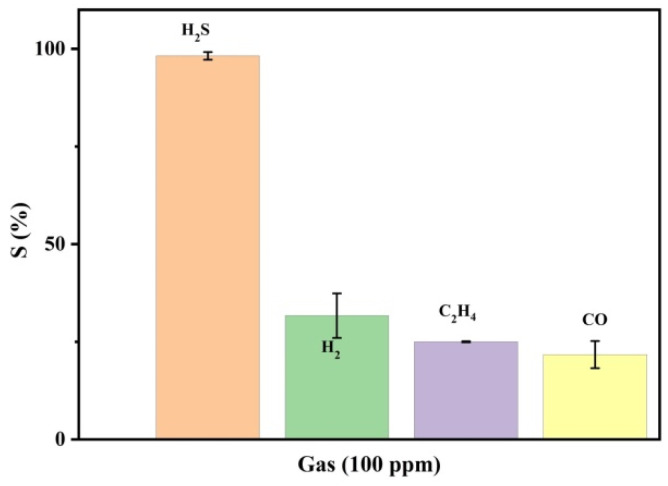
Selectivity of the Cu-MOF/PVA/IL membrane to H_2_S in comparison to H_2_, C_2_H_4_, CO gases at 100 ppm at room temperature.

**Figure 9 nanomaterials-12-00913-f009:**
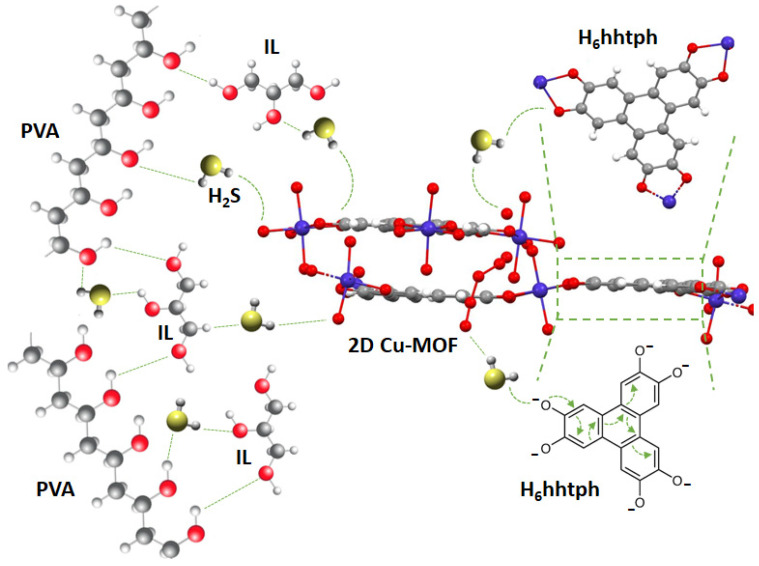
Illustration of the suggested H_2_S gas sensing mechanism. All atoms are color-coded; Cu: purple, O: red, C: grey, H: white, S: yellow.

**Table 1 nanomaterials-12-00913-t001:** Sensor performance comparison with literature-reported values.

Sensor/Material	Optimum Operating Temperature (°C)	Gas	Detection Limit	Refs.
Cu_3_(HHTP)_2_/PVA/IL	23 °C	H_2_S	1 ppm	This Work
Cu/Ni (HHTP)	RT	H_2_S/NO	40–80 ppm	[26]
Cu_3_(HHTP)_2_	250 °C	Acetone	50 ppm	[28]
		H_2_/CO	20 ppm	
		CO_2_/NH_3_	200 ppm	
		CH_4_	500 ppm	
		NO_2_	1 ppm	
PVA-Semiconducting nanoparticles (CuO, ZnFe_2_O_4_, CuFe_2_O_4_, and WO_3_) based sensors	80 °C	H_2_S	10 ppm	[32,33]
Pd and Pt NPs doped Cu_3_(HHTP)_2_	RT	NO_2_	1 ppm	[42]

## Data Availability

Not applicable.

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
