# Peer review of "Flexible Cu3(HHTP)2 MOF Membranes for Gas Sensing Application at Room Temperature"

_nanomaterials, 2022, doi:10.3390/nano12060913_

Round 1
Reviewer 1 Report
This paper is organized very well and I enjoy it and I believe it can be published with some minor corrections.
1) I think it is better the authors give detail for fabrication, measurement, and material process some steps to it becomes clear for readers of this paper in their research, it can increase the attention and citation of this paper in future
2) the quality of the image should be improved
3) it is not clear which software is used for simulation and condition of the simulation
4) why did you select 23C as the test of H2S and why you didn't check other temperatures as you should in comparison table such as 80
5)it is not clear how did you make the decision to use Cu-MOF/PVA/IL as the final coat of your sensor
6) can you give some physical or chemical reason why the other gas shows lower S in comparison with H2S, may be junction between gas and membrane is the reason
Author Response
Response to the Reviewers’ Comments
The authors are grateful for the reviewers’ valuable comments and suggestions that have improved the quality of this manuscript.
Reviewer # 1
This paper is organized very well, and I enjoy it and I believe it can be published with some minor corrections.
1) I think it is better the authors give detail for fabrication, measurement, and material process some steps to it becomes clear for readers of this paper in their research, it can increase the attention and citation of this paper in future
Response: Done. Details of the sensor’s fabrication and measurements have been included in the revised manuscript.
2) the quality of the image should be improved
Response: Done. The required graphs have been replaced with higher resolution images.
3) it is not clear which software is used for simulation and condition of the simulation
Response: Chemdraw software was used to simulate the structures using the data obtained from the PXRD patterns. The PXRD pattern for comparison was simulated using Highscore Plus software using the lattice parameters from the reported literature. The revised manuscript has been updated with the above information.
4) why did you select 23C as the test of H2S and why you didn't check other temperatures as you should in comparison table such as 80
Response: The temperature recorded in our lab while doing the tests is 23°C. Therefore, it is the room temperature (RT). Moreover, our main objective has been to develop sensors that operate at RT in order to reduce the power consumption and the cost of the device. In fact, this reflects the novelty of our work compared with other sensors and hence no need to heat the sensor at higher temperatures and consume more energy.
5)it is not clear how did you make the decision to use Cu-MOF/PVA/IL as the final coat of your sensor
Response: It is well known in the literature that copper has high affinity towards H2S gas, and several research works have been carried out to sense the H2S gas using copper particles with different nanostructures. But those sensors are either working at high temperatures or their detection limit is not high enough for these applications. To date, Cu-MOF membrane has not been used to sense H2S gas. Moreover, MOF materials are characterized by their high porosity and high surface area. Hence these features are expected to provide a potential for gas absorption with high sensitivity of the sensor fabricated thereafter. Thus, we proposed to conduct this research to increase the detection limit up to 1ppm and develop a high-performance sensor at room temperature.
6) can you give some physical or chemical reason why the other gas shows lower S (response) in comparison with H2S, may be junction between gas and membrane is the reason
Response: The fabricated composite membrane comprises highly functionalized components, in which the formation of an extensive H-bonding throughout the composite membrane has been proven. Unlike other gases mentioned in this study, the slightly polar H2S gas molecules contribute to the H-bonded network through the attraction of the H atoms of H2S to the highly electronegative oxygen atoms that are speared over all components of the composite membrane. Moreover, the presence of Cu2+ nodes (borderline acid) of Cu3(HHTP)2 facilitate adsorption of H2S molecules through interaction with S atom. Though, the available groups do not show such affinity towards the other gases tested in this work and hence showed lower responses.
Reviewer 2 Report
Flexible Cu3(HHTP)2 MOF Membranes for Gas Sensing application at Room Temperature by Ali et al. discussed the synthesis and characterization of Cu3(HHTP)2 MOF Membrane for sensing H2S at the room temperature. The research article was interesting specifically from the perspective of its application on measuring gases such as H2S. However, it suffered from various flaws. Some of the major flaws are as follows:
I am not sure if the manuscript was placed into the journal’s template because I could not see the numbering of the lines/sentences that would be easier to point out.
The overall grammar of the manuscript was very weak and did not convey well. There were bad instances of grammar throughout the manuscript. The sentences jumped back and forth from past to present tense. Some sentences failed to convey their full meaning, for example, the first sentence “Severe environmental impact…” had grammatical errors, then the same paragraph sentence stating, “one among them is….”
This whole paragraph was confusing and had bad instances of grammar. Then the last sentence of the same paragraph, “some result in non-lethal effect………”I am not sure why pollution is being discussed here. We are all aware of the consequences of pollution. The introduction needs to be rewritten with a better hold of sentences and clear language. The second paragraph also had similar grammatical issues.
Page 12 second paragraph, acetone was referred to as “GAS”. ACETONE IS A LIQUID. It was very disappointing to see so many mistakes in this poorly written manuscript.
Also, there were so many grammatical errors in the last paragraph of the introduction (specifically, the first sentence and the last sentence).
The overall quality of the introduction was very poor.
The material section was incomplete. There was no discussion on IL (ionic liquids) which one, why, and what ratios. This all should be included in the material section
In synthesis 2.2 and section 2.3, the use of DI and DISTILLED water was jumping back and forth. Please ensure which one was used? If DI was it produced in the laboratory? How? Which equipment?
The amounts of the substances were somewhere in mmoles and somewhere in grams? Why?
The instrumentation details should have been provided. Only by providing one reference is not enough.
In my opinion, in addition to the PXRD, SEM, and FT-IR studies, the complete morphology, microstructure, and elementary composition of the sample should have been reported e.g., by X-ray and photoelectron spectroscopy (XPS), and transmission electron microscope (TEM). For newly synthesized materials the complete characterization is possible based on all of these techniques. Please consider including XPS and TEM.
Figure 6 appeared to be blurred. It was hard to read.
In my opinion, the manuscript needs to be re-written and without bad instances of English. Furthermore, the materials, methods, and instrumentation section should have all details. If all details are not possible in the manuscript then should be added as supplementary files. Furthermore, more characterization should be included for a higher quality of presentation such as XPS and TEM (please see above).
Author Response
Response to the Reviewers’ Comments
The authors are grateful for the reviewers’ valuable comments and suggestions that have improved the quality of this manuscript.
Reviewer # 2
1) Flexible Cu3(HHTP)2 MOF Membranes for Gas Sensing application at Room Temperature by Ali et al. discussed the synthesis and characterization of Cu3(HHTP)2 MOF Membrane for sensing H2S at the room temperature. The research article was interesting specifically from the perspective of its application on measuring gases such as H2S. However, it suffered from various flaws. Some of the major flaws are as follows:
I am not sure if the manuscript was placed into the journal’s template because I could not see the numbering of the lines/sentences that would be easier to point out.
Response: In fact, the manuscript was not drafted in the journal’s template, but it is written in normal format and submitted to the journal because the journal’s guidelines said “your manuscript your format” hence the line number were not included into the first draft. The revised manuscript was formatted by the journal’s Editorial office into the current format, which is now in accordance with the comment of the reviewer.
2) The overall grammar of the manuscript was very weak and did not convey well. There were bad instances of grammar throughout the manuscript. The sentences jumped back and forth from past to present tense. Some sentences failed to convey their full meaning, for example, the first sentence “Severe environmental impact…” had grammatical errors, then the same paragraph sentence stating, “one among them is….”
This whole paragraph was confusing and had bad instances of grammar. Then the last sentence of the same paragraph, “some result in non-lethal effect………”I am not sure why pollution is being discussed here. We are all aware of the consequences of pollution. The introduction needs to be rewritten with a better hold of sentences and clear language. The second paragraph also had similar grammatical issues.
Response: Thanks for the comment. The manuscript has been revised to fix the grammar, typos, and unclear sentences.
It is quite right that the scientific community is aware of the consequences of pollution and hence the researchers are working to minimize its effect on human health. Thus, this work is part of those efforts to fabricate highly sensitive and selective sensors to detect of one those pollutants gases such as H2S at very low concentrations “1ppm”. Hence, highlighting the effect of pollution on human health is important for the public and interested people who want to work in this vital area of research.
3) Page 12 second paragraph, acetone was referred to as “GAS”. ACETONE IS A LIQUID. It was very disappointing to see so many mistakes in this poorly written manuscript.
Response : Acetone is indeed in liquid phase and must be vaporized to detect few ppm in gas phase as reported in the literature [Ref. 27 and 28]. Therefore, in the reported work it is referred to as a gas. Furthermore, a correction has been made to emphasize the gas phase of acetone, where it is now mentioned as “acetone vapor”
4) Also, there were so many grammatical errors in the last paragraph of the introduction (specifically, the first sentence and the last sentence). The overall quality of the introduction was very poor.
Response: The errors have been corrected in the revised manuscript as advised.
5) The material section was incomplete. There was no discussion on IL (ionic liquids) which one, why, and what ratios. This all should be included in the material section
Response: The electrical conductivity of the PVA films is controlled by mixing them with suitable concentrations of an ionic liquid (IL) such as glycerol. Ionic liquids (ILs) are well known for their good ionic conductivity that is attained at room temperature. In general, ILs serve as electrolytes and diffusion barriers, and they have low values of vapor pressure, and low toxicity. 5 vol% of IL (glycerol) was mixed in 20 ml of the PVA stock solution and doped with Cu-MOF suspended in 2.5 ml of DW DI at 70°C at 1100 RPM. This statement has been added to the material section 2.3 (page 3, lines 14-18).
6) In synthesis 2.2 and section 2.3, the use of DI and DISTILLED water was jumping back and forth. Please ensure which one was used? If DI was it produced in the laboratory? How? Which equipment?
Response: Distilled water produced in the lab using the Milli-Q system from Elix Technology was used throughout the experiments. This clarification was also added to the experimental section of the manuscript. Distilled water has been referred to as DW throughout the study. Other relevant errors have been rectified in the revised manuscript.
7) The amounts of the substances were somewhere in mmoles and somewhere in grams? Why?
Response: The amounts of substance that have been mentioned as mmoles are instances where the raw material was used to prepare the Cu-MOF. After the MOF has been synthesized as powders, the amount of the powders used to fabricate the membrane were measured in grams.
8) The instrumentation details should have been provided. Only by providing one reference is not enough.
Response: The instrument details have been included into the revised manuscript as suggested.
9) In my opinion, in addition to the PXRD, SEM, and FT-IR studies, the complete morphology, microstructure, and elementary composition of the sample should have been reported e.g., by X-ray and photoelectron spectroscopy (XPS), and transmission electron microscope (TEM). For newly synthesized materials the complete characterization is possible based on all of these techniques. Please consider including XPS and TEM.
Response: We do agree that the XPS and TEM would provide more data in support to the morphology studies but unfortunately, it would not be possible at this stage for the following reasons:
- Our main aim is to fabricate the sensor and study it’s performance for electronic devices applications.
- Based on the SEM micrograph of the as-synthesized Cu-MOF (Fig 3A), it is evident that Cu-MOF exhibits a 2D flakes morphology with dimensions at the mm scale. Therefore, the authors believe that NO useful information could be obtained using TEM analysis of these MOF crystallites.
- Moreover, the time given by the journal to address the reviewers’ comments is only 8 days. However, the following TEM image is obtained and shows agglomerated MOF particles. We could obtain a better TEM image if we have given more time. Unfortunately, we do not have XPS machine at our university.
Fig R1: TEM Image of the Cu-MOF particle
10) Figure 6 appeared to be blurred. It was hard to read.
Response: The figure has been divided into two figures Fig. 6A and Fig.6B to become very clear. Fig. 6A shows the response of the sensor with respect to time at different H2S gas concentrations while Fig. 6B displays the response of the sensor as a function of different H2S gas concentrations.

Round 2
Reviewer 2 Report
The manuscript has been revised appropriately. However (in response to the point no 5) I would suggest adding a few appropriate references of using glycerol as an ionic liquid/co-solvent and the reason why it was chosen for this experiment.
Author Response
Response to the Reviewer’s Comment
The authors are grateful for the reviewer’s comment and suggestion that have improved the quality of this manuscript.
Reviewer # 1
The manuscript has been revised appropriately. However (in response to the point no 5) I would suggest adding a few appropriate references of using glycerol as an ionic liquid/co-solvent and the reason why it was chosen for this experiment.
Response: Although we have addressed this point in the previous response to the reviewer’s comment, we will elaborate more and add more references in this version of the revised manuscript. The following paragraph has been added into page 3, section 2.3 of the revised manuscript.
The polymer membrane serves as a host for the MOFs microparticles. The electrical conductivity of a PVA membrane can be controlled by doping with a suitable ionic liquid such as sorbitol, 1-methyl-3-n-decyl-imidazolium bromide, and glycerol [35-37]. Ionic liquids (ILs) are well known for their good ionic conductivity that is attained at room temperature. Mainly, ILs serve as electrolytes and diffusion barriers, and they have low values of vapor pressure, and low toxicity. Moreover, ILs are suitable to be used for the fabrication of electrochemical sensors as they are environmentally friendly [32,33]. It has been shown that glycerol-IL can be used effectively to control the conductivity of PVA membranes [37,38], thus, it will be utilized in this work.
